# Quantifying the variability of the annular modes: Reanalysis uncertainty versus sampling uncertainty

Edwin P. Gerber[1] and Patrick Martineau[2]

[1]Courant Institute of Mathematical Sciences, New York University, 251 Mercer Street, New York NY 10012, USA
[2]Research Center for Advanced Science and Technology, University of Tokyo, Japan

**Correspondence:** Edwin P. Gerber (gerber@cims.nyu.edu)

**Abstract.** The annular modes characterize the dominant variability of the extratropical circulation in each hemisphere, quantifying vacillations in the position of the tropospheric jet streams and the strength of the stratospheric polar vortices. Their representation in all available reanalysis products is assessed. *Reanalysis uncertainty* associated with limitations in the ability to constrain the circulation with available observations, i.e., the inter-reanalysis spread, is contrasted with *sampling uncertainty*
associated with the finite length of the reanalysis records.

It is shown that the annular modes are extremely consistent across all modern reanalyses during the satellite era (c. 1979 onward). Consequently, uncertainty in annular mode variability, e.g., the coupling between the stratosphere and troposphere and the variation in the amplitude and time scale of jet variations throughout the annual cycle, is dominated by sampling uncertainty. Comparison of reanalyses based on conventional (i.e., non-satellite) or surface observations alone with those using
all available observations indicates that there is limited ability to characterize the Southern Annular Mode (SAM) in the presatellite era. Notably, prior to 1979, surface-input reanalyses better capture the SAM at near surface levels than full-input reanalyses. For the Northern Annular Mode, however, there is evidence that conventional observations are sufficient, at least from 1958 onward. The addition of two additional decades of records substantially reduces sampling uncertainty in several key measures of annular mode variability, demonstrating the value of more historic reanalyses. Implications for the assessment of
atmospheric models and the strength of coupling between the surface and upper atmosphere are discussed.

## 1 Introduction

The annular modes characterize the dominant internal variability of the extratropical atmosphere (Thompson and Wallace,
2000). In the troposphere, they primarily characterize meridional shifts in the extratropical jet stream, a positive index indicating a poleward shift. The jet stream is associated with the extratropical storm tracks, such that the annular mode indices indicate shifts in storm activity, particularly in Northern Europe and eastern North America (e.g., Thompson and Wallace, 1998). In the

stratosphere, the annular modes chiefly characterize variations in the strength of the polar vortex, a positive index indicating a stronger than average vortex. Baldwin and Dunkerton (2001) revealed a downward influence of the stratospheric polar vortex on the tropospheric jet stream by comparing annular mode indices computed at different pressure levels.

In this study, we focus on the representation of the annular mode indices, or time series, in all available global reanalysis products. The spatial structure of the annular modes is characterized by a meridional dipole, where mass (or equivalently, momentum, as the large scale flow is geostrophically balanced) is exchanged between the high and low latitudes (Thompson and Wallace, 2000). It can be understood as the gravest mode of a varying jet (e.g., Vallis et al., 2004; Gerber and Vallis, 2005) and is well captured across a range of models (e.g., Gerber et al., 2010). The term "annular mode" was derived from the annular structure of the patterns in longitude, where geopotential height varies in the same phase at a given latitude (Limpasuvan and Hartmann, 2000). As discussed by Deser (2000) and Ambaum et al. (2001), among others, such coherent fluctuations are generally not observed, and the annular structure is primarily statistical in nature (Gerber and Thompson, 2017). The temporal variability of the annular modes provides a convenient means to quantify coupling between tropospheric jets and the stratospheric vortices, and variations in the amplitude and persistence of the variability through the annual cycle. It has proven more challenging to understand and model, and is thus the focus of this study.

The jet streams are known to vacillate on slower time scales than individual weather systems, as first quantified with the "zonal index" (e.g., Rossby, 1939; Namias, 1950). A number of studies have investigated how this enhanced persistence may result from a feedback between eddies and the mean flow (e.g., Robinson, 1996; Feldstein and Lee, 1998; Lorenz and Hartmann, 2001; Gerber and Vallis, 2007; Barnes et al., 2010; Zurita-Gotor et al., 2014). The stratospheric polar vortex exhibits greater memory than the troposphere, with implications for predictability of the tropospheric jet stream on sub-seasonal to seasonal time scales (e.g., Baldwin et al., 2003; Sigmond et al., 2013). Models, both idealized (e.g., Gerber et al., 2008b) and comprehensive (e.g., Gerber et al., 2008a), tend to exhibit too much persistence, with possible implications for their projected response to global warming (Kidston and Gerber, 2010; Barnes and Hartmann, 2010; Son et al., 2010). It is thus of both practical and theoretical interest to determine the limits to which we can quantify the temporal variability of the annular modes of our atmosphere with reanalyses.

After briefly describing the available reanalyses and our procedure for evaluating the annular modes in Sections 2 and 3, we explore annular mode variability in the most recent products of the four major reanalysis centers in Section 4, establishing a "Reanalysis Ensemble Mean" as a standard for comparison with all reanalysis products. In sections 5 and 6, we compare the reanalyses in the satellite and pre-satellite eras, respectively. We find that the Northern Annular Mode (NAM) can be well constrained with conventional observations, and appears consistent across a number of reanalyses since 1958. Characterization of Southern Annular Mode (SAM) variability, however, appears to depend more strongly on satellite observations, and the SAM index varies substantially between reanalyses before 1979. There is evidence, however, that surface-input reanalyses can capture the tropospheric SAM, at least on monthly time scales, as far back as the late 1950s. In Section 7, we further explore historic reanalyses constrained only by surface measurements, and find that there is potential to constrain the NAM at tropospheric levels in earlier periods.

In Section 8 we show that sampling uncertainty is the leading source of uncertainty in our ability to constrain the temporal variability of the annular modes. We focus in particular on the downward coupling associated with extreme vortex variability and the time scales of annular mode variability as a function of height and season. Our conclusions are summarized in Section 9.

## 2   The reanalyses

Table 1 lists the reanalyses compared in this study. Fujiwara et al. (2017) provide a detailed explanation of the data assimilation procedures and input data sets that differentiate these reanalysis products. Here, we highlight a few key differences that are pertinent to this study. ERA-Interim, JRA-55, MERRA2, and CFSR/CFSv2 represent the current state-of-the-art reanalysis products of the four major reanalysis centers. While this is an moving target, e.g., the new ERA5 reanalysis will soon replace ERA-I as the top product available from the European Center for Medium Range Weather Forecasting (ECMWF), a key finding of this study is that all of the modern reanalyses accurately capture the variability of the annular modes.

In constructing a *re*analysis, one seeks a balance between the aim of providing a best possible estimate of the atmospheric state at any given time (as with an *analysis* for providing initial conditions for a weather forecast), with the goal of providing a homogeneous record of the atmospheric evolution. The latter requirement generally entails a choice of a fixed atmospheric model and assimilation procedure for the entire reanalysis. This is an issue with the CFSR and CFSv2 reanalyses. The National Center for Environmental Prediction (NCEP) made substantial upgrades to their reanalysis system as of Jan. 1 2011, including a change in the resolution of the atmospheric model as documented in Saha et al. (2014). We find that the annular modes appear to be continuous and consistent across this break, but it is best practice to consider CFSR and CFSv2 as separate reanalyses.

A more difficult decision concerns the use of new observations that become available during the record. A most notable change is the introduction of satellite measurements around 1979. Most reanalyses make use of all observations as they become available. This provides an increasingly accurate representation of the atmospheric state with time, but introduces the potential for spurious trends which are not associated with physical changes in the atmosphere, but rather an enhanced ability to observe it. A good example of these artificial trends can be seen in stratospheric temperatures, as shown in Fig. 1 of Long et al. (2017). In earlier reanalyses, global mean temperatures jump abruptly in response to new satellite measurements. More recent reanalyses have adopted bias correction techniques to adjust for the errors in earlier measurements, but there is still potential for spurious trends over time.

A few reanalyses explicitly limit the observational inputs, giving us an opportunity to assess the impact of new observations. In particular, ERA-20C and NOAA-20CR v2 and v2c intentionally limit themselves to surface based observations only; we refer to these as "surface-input reanalyses". This choice allows them to provide a more uniform estimate of the atmospheric state as far back as 1850. They are not immune to spurious trends, however, as the availability of surface measurements fluctuations with time.

The JRA-55C reanalysis uses the same model and data assimilation procedure as in JRA-55, but explicitly excludes all satellite measurements. We refer to it as a "conventional-input" reanalysis, acknowledging that all reanalyses are, by this

definition, conventional-input before satellites became available. We use 1979 as a the start of the satellite-era, as this is the first full year for which satellite measurements were available continuously, but note that a few satellite measurements were available earlier. JRA-55C provides an opportunity to assess the impact of satellite measurements on our ability to quantify the annular modes. Finally, we consider the JRA-55AMIP integration, which is not a reanalysis. It only considered observed sea

surface temperatures as an observational input, i.e., it is the output of a free running atmospheric model used in the JRA-55 and 55C reanalyses, constrained only by the sea surface temperature and observed changes in radiative forcing. As will be shown, it allows us to confirm the conventional wisdom that in general, the extratropical circulation is only weakly constrained by sea surface temperature.

## 3   The annular modes

As suggested by Baldwin and Thompson (2009), we use a simplified procedure for computing the annular mode based on the polar cap averaged geopotential height, where the cap is defined as all latitudes poleward of $65°$. Following Gerber et al. (2010), we first remove the global mean geopotential height at each time step. This focusses the analysis on meridional variations in geopotential height (i.e., shifts in mass), as opposed to systematic variations in the entire layer associated with changes in atmospheric temperature. In practice this only has a significant impact on the indices in the upper stratosphere and above, but

it also helps removes trends associated with global warming. Based on model integrations (e.g., Gerber et al., 2010), we expect that trends will be most significant near the tropopause, where there is a sharp gradient in the character of the annular mode as it shifts from characterizing the tropospheric jet to the stratospheric vortex.

Fig. 1 illustrates our procedure, which we spell out precisely in the following.

1. 6 hourly output of geopotential height is averaged to form a daily time series $Z(t, \lambda, \phi, p)$, where $\lambda$, $\phi$, and $p$ refer to
latitude, longitude, and pressure. For leap years, the 366 days are interpolated to 365, but we note the impact of this interpolation is inconsequential, and one could more simply remove one day from each leap year.

2. The annual cycle is formed by averaging each calendar day over the record, and then smoothing the resulting annual cycle with a 60 day low pass Lanczos digital filter. The anomalous height $Z'(t, \lambda, \phi, p)$ is formed by removing the smoothed annual cycle. The daily, zonal mean anomalous height is illustrated for a single year and pressure level in Fig. 1a.

3. We then compute the global mean geopotential height, $\overline{Z'}^{global}$, and austral and boreal polar cap heights, $\overline{Z'}^{SH}$ and $\overline{Z'}^{NH}$, as illustrated in Fig. 1b.

4. Raw SAM and NAM time series are defined by $-(\overline{Z'}^{SH} - \overline{Z'}^{global})$ and $-(\overline{Z'}^{NH} - \overline{Z'}^{global})$, respectively. The negative sign is in keeping with the standard convention of the annular mode introduced by Thompson and Wallace (1998).

5. The standardized SAM and NAM indices are obtained by dividing the raw indices by their standard deviations, yielding
indices with unit variance by construction, as illustrated in Fig. 1c.

As steps 1-4 are linear, one can rearrange them to minimize the computational effort. In practice, we first compute the global and polar cap average geopotential heights, and then daily average and deseasonalize them to form time series $\overline{Z'}^{global}$, $\overline{Z'}^{SH}$, and $\overline{Z'}^{NH}$.

While this simple definition of the annular mode index appears to depend only on geopotential height over the pole, we stress that it captures variability throughout the midlatitudes. Fig. 2 illustrates the latitudinal structure of the annular mode variability, obtained by regressing daily geopotential height anomalies on a given level on the respective annular mode index, i.e., the 850 hPa patterns are obtained using the 850 NAM and SAM indices, and so forth. It compares remarkably well with the structure obtained in EOF calculations (e.g. Gerber et al., 2010, their Figure 4). The broad latitudinal extent of the patterns reflects the strong anticorrelation between geopotential height in the high and low latitudes, as necessitated by the conservation of mass (cf. Gerber and Vallis, 2005). One can even obtain the zonal structure of the annular mode patterns by regressing two dimensional (latitude-longitude) geopotential on the annular mode index (not shown): in the Northern Hemisphere, the zonal mean geopotential anomalies are dominated by localized variability in the Atlantic and Pacific storm tracks.

By convention, the AM indices are normalized to have unit variance. The patterns thus capture the amplitude of variations. In the troposphere, variability is stronger in the austral hemisphere, as seen in the increased amplitude of the SAM relative to the NAM at 850 hPa. In the stratosphere, however, variability is stronger in the boreal hemisphere, as seen with the 100 and 10 hPa patterns.

We took the opportunity in Fig. 2 to contrast the patterns obtained from a full-input reanalysis (ERA-Interim, which is characteristic of the other modern reanalyses) with those in a surface-input reanalysis (ERA-20C). The agreement is weaker at higher altitudes, consistent with a decorrelation between the indices at height (see Section 5), but we find it notable that a reanalysis which only assimilates surface information can reasonably capture the structure of annular mode variability at 10 hPa. This is in part a reflection of the fact that the dipolar structure of the annular modes is fairly generic, and well obtained by any free running atmospheric model.

The formation of the annual cycle and normalization of the time series (steps 2 and 5) depend on the length of the record. If one follows this procedure to define the annular mode over two distinct periods that overlap, the resulting indices will not agree perfectly during the period of overlap. We find that that difference is inconsequential provided one uses a period of sufficient length, a decade or so in practice. This limitation could be addressed by defining the climatology and normalization constants over a set period, for instance the World Meteorological Organization (WMO) climatology spanning the last three full decades. For the purposes of this paper, we have computed the indices separately for each period of comparison, e.g., 1981-2010 for our comparison of the reanalyses in Sections 4 and 5, and 1958-1978 for our comparison of reanalyses in the pre-satellite era in Section 6.

We focus on a subset of the pressure levels that were shared by all reanalyses: 1000, 850, 700, 600, 500, 400, 300, 250, 200, 150, 100, 70, 50, 30, 20, 10, 7, 5, 3, 2, and 1 hPa. Levels above 10 hPa are unavailable for NCEP-R1/R2 and NOAA 20CR v2/v2c reanalyses. The annular mode indices are fairly uniform within the troposphere and stratosphere, respectively, and such fine vertical resolution is really only needed in the tropopause region.

Lastly, we remark that our definition of the annular mode (or equivalently, previous Empirical Orthogonal Function based definitions) require extrapolation of data below the surface. This was done by the reanalysis centers for all reanalyses with the exception of the MERRA products. We have opted to omit MERRA and MERRA2 from comparisons below 700 hPa, where their data was incomplete.

## 4   Consistency in the representation of the annular modes in state-of-the-art reanalyses

We first compare the four modern reanalyses, ERA-Interim, JRA-55, MERRA2, and NCEP-CFSR, to justify the use of "Reanalysis Ensemble Mean" (REM) annular mode indices as a benchmark of comparison. We focus on the period 1981-2010, which corresponds to the standard 30 year climatological evaluation period by WMO convention. This also conveniently avoids the break between the CFSR and CFSv2 reanalyses, but we assess this transition below, in addition to preliminary output from the new ERA5 reanalysis.

Fig. 3 shows the pairwise squared correlation ($R^2$) of the NAM and SAM indices as a function of pressure. The squared correlation coefficient indicates the fraction of variance shared by two time series. We find that the modern reanalyses share approximately 96% of the variance or greater at all levels in both hemispheres, with the exception of the upper stratosphere in the austral hemisphere.

In the Northern Hemisphere, the agreement is 99% or better between ERA-Interim, JRA-55, and MERRA2, with slightly weaker correlation to CFSR in the upper troposphere and lower stratosphere, where there is a minimum in correlation at 200 hPa near the extratropical tropopause. CFSR, however, does not stand out from the other reanalyses at this level in the austral hemisphere. There is a tendency towards better agreement in the lower and mid-stratosphere as compared to the troposphere in both hemispheres, but we stress that the correlation is always extremely high.

Given the strong agreement of all the modern reanalyses, we adopted a REM annular mode index based on the average of ERA-Interim, JRA-55, and CFSR. MERRA2 was omitted from the REM due to missing data at lower levels, but the results are nearly identical if it is included. We emphasize that this decision does not imply a value judgement of the quality of MERRA2 reanalysis.

The intensity of annular mode variability changes throughout the annual cycle, particularly in the stratosphere (e.g., Baldwin et al., 2003; Gerber et al., 2010). This can be seen in the annual cycle of annular mode variance in Fig. 4 (black contours). In the troposphere, the annular modes are most variable in the winter seasons in both hemispheres, with a greater variation over the annual cycle in the boreal hemisphere. In the stratosphere, there is considerably more variation throughout the annual cycle: the maximum is approximately 4 times the annual mean, and variability drops to near zero in the summer. There is also a notable difference in the timing of peak variability between the two hemispheres. For the NAM, the maximum is collocated with the maximum in tropospheric variability in the winter, while for the SAM, variability is greatest in late spring.

Given this variation of the annular modes within the annual cycle, we check the consistency between the reanalyses as a function of day of year, illustrated by the shading in Fig. 4. We find that the indices agree well ($R^2 > 0.95$) at all levels and

heights except when the variability is weak during the summer, where on average, the pairwise $R^2$ drops as low as 0.5. Thus, there is great certainty in the annular mode state except in periods when the annular mode is inconsequential.

Finally, we consider the consistency of the annular mode indices in more recent years. NCEP made substantial upgrades to their reanalysis system starting in 2011, replacing the CFSR reanalysis with CFSv2. To assess the new system, in Fig. 5 we compare the $R^2$ correlation between the SAM in CFSR with other reanalyses between 2005-2010 with the same metrics for CFSv2 over the period 2011-2016. This choice of periods provided the most fair comparison: 6 years for each reanalysis. We find that there is a reduction in the spread between CFSv2 and the other reanalyses relative to CFSR, most notably at upper levels, where the $R^2$ correlation increases to approximately 99% from 95%. As seen in Fig. 5b, however, coherence is increased at all levels – although it is important to point out that CFSR was already highly correlated with the other reanalyses. We have not shown the equivalent plots for the NAM because we found no discernable difference between the two periods; both plots were equivalent to Fig. 3a. This is in part because the NAM in CFSR was already highly correlated at all levels and their was more room for improvement with the SAM. We have additionally inspected the annular indices across the break from CFSR and CFSV2 at 31 December 2010, and found no abrupt changes (not shown).

ECMWF is in the process of producing a new reanalysis, ERA5. We have found ERA5 to be virtually indistinguishable from the other modern reanalyses over the period for which it is currently available, 2008-2016. During this interval, the NAM and SAM based on ERA5 output are $R^2$ correlated at 0.998 or better with those in ERA-Interim and MERRA2 at all available levels. For JRA-55, $R^2$ exceeds 0.996 (0.991) for the NAM (SAM); the SAM coherences is only less than 0.996 near the surface, where interpolation is likely the source of error. Care must be taken in comparing with CFSR/CFSv2, given the transition at the end of 2010, and Fig. 5 shows the high correlation with ERA5 over the period 2011-2016. ERA5 should ultimately be available from 1950 onward, and it will be interesting to explore the pre-satellite era in this reanalysis.

## 5 Comparing the representation of the annular mode indices across all reanalyses

Fig. 6 shows the $R^2$ correlation between the annular mode indices computed from each individual reanalysis with the REM annular mode index over the period 1981-2010. This period provides nearly optimal overlap between the reanalyses: all but one (ERA-40) are available the entire period. The correlation for ERA-40 was based on the years when it was fully available, 1981-2001. We enumerate the key findings below.

*(1) The NAM is well captured throughout the troposphere and the bulk of the stratosphere by all reanalyses that assimilate free atmospheric data.* The $R^2$ correlation with the REM exceeds 0.975 from the surface to 10 hPa for all reanalyses excepting ERA-20C, JRA55AMIP, and NOAA 20CR v2/2c. The modern reanalyses are somewhat better correlated with the REM than earlier generation products, particularly in the lower to mid-stratosphere. While this could partly arise by construction (the REM is based on the modern reanalyses), MERRA and MERRA2 are not a part of the REM and still exhibit enhanced correlation. MERRA appears slightly more highly correlated than MERRA2, although the difference is trivial.

*(2) The SAM is well captured by most reanalyses that assimilate satellite measurements, and is demonstratively better represented in the more recent reanalyses.* Most of the comprehensive reanalyses are well correlated with the REM ($R^2 > 0.92$

up to 10 hPa), but in comparison to the Northern Hemisphere, the modern reanalyses are consistently better than earlier reanalyses; MERRA and MERRA2 are better correlated with the REM that the earlier reanalyses (ERA-40, JRA-25, NCEP-R1 and R2) at all levels.

*(3) Satellite measurements are critical for the representation of the SAM, but not for the NAM.* The representation of the NAM by JRA-55C, which assimilates only conventional observations, is nearly indistinguishable from the second generation reanalyses up to 30 hPa, and is still very good up to 10 hPa. We conclude that conventional observations are sufficient until the upper stratosphere, which suggests that there is potential for a skillful representation of the NAM in the pre-satellite period, as further investigated in Section 6. In the austral hemisphere, however, the representation of SAM in JRA-55C is demonstrably poorer at all levels (though it still captures more than 85% of the variance up to 3 hPa). JRA-55C is most strongly correlated with the other reanalyses in the mid-stratosphere, but decays considerably towards the surface, where it only captures 85% of the variance, substantially less than the reanalyses which assimilate only surface measurements.

*(4) Reanalyses based on surface measurements alone remain highly correlated with the REM in the troposphere, but the correlation falls off sharply above the tropopause, particularly for the 20CR reanalyses.* ERA-20C, however, remains remarkably well correlated through the stratosphere, capturing more than half the variance of the REM up to the stratopause at 1 hPa. As discussed in greater detail in Section 7, this indicates that the stratosphere is sufficiently coupled to the surface to extract useful information about its state from surface data alone.

We finally note that sea surface temperatures are insufficient to capture AM variability, at least in the case of the JRA-55AMIP integration. It has long been known that the extratropical circulation is not driven by local SSTs (e.g., Barsugli and Battisti, 1998). Tropical SSTs are known to influence the annular modes (e.g., El Niño; L'Heureux and Thompson, 2006), but appear insufficient to constrain the annular mode variability of either hemisphere.

## 6   Pre-satellite representation of the annular modes

The ability of the JRA-55C to accurately capture the NAM without the aid of satellite measurements suggests that it should be possible to capture it during the pre-satellite era, at least with full-input reanalyses that make use of all available observations. Fig. 7 compares the 6 available reanalyses during the period 1958-1978. As it is less trivial to identify a meaningful REM during this period, we consider two comparisons against a given individual reanalysis, JRA-55 and NCEP-R1, respectively. Comparable results are found when using ERA-40 as a reference, and the selection here is not meant to be a value judgment. Based on Fig. 7a and b, we concluded that the NAM is consistently represented throughout the troposphere and stratosphere in ERA-40, JRA-55, and NCEP-R1, giving us confidence that there are sufficient observations to quantify the NAM as far back as 1958.

While the NAM in the surface-input reanalyses remains highly correlated in the troposphere, we note that ERA-20C is not as well correlated in the stratosphere as it was in more recent years. At 10 hPA, ERA-20C only captures about 40% of the variance, as compared to 60% between 1981 to 2010.

The situation is quite different in the austral hemisphere, with widespread divergence between the reanalyses (Fig. 7c,d). JRA-55 shares only 30% of the variance with the other reanalyses throughout the troposphere. NCEP-R1 is more strongly correlated with the NOAA 20CR and ERA-20C reanalyses in the troposphere (sharing approximately 60% of the variance), but poorly correlated with the other full reanalyses (JRA-55 and ERA-40). This is partly expected, given the limited ability of JRA-55C to capture the SAM during the satellite era. JRA-55C was still fairly well correlated with the REM in the satellite era, however, suggesting that the poor representation of the SAM before 1979 also reflects a dearth of conventional observations.

The poor correlations in Fig. 7(c,d) begs the question: do any of the reanalyses have skill before 1979? It is difficult to characterize the synoptic evolution of the SAM from direct measurements, but Marshall (2003) developed a station-based index to track the SAM at the surface on monthly time scales. Briefly, the index uses a series of stations to estimate the difference in zonal mean pressure between 65 and 40°S. Table 2 compares correlation between the Marshall (2003) index over the pre- and post-satellite periods with the 850 SAM index. From 1979 and 2001, the station-based index is approximately 0.85 correlated with the near-surface SAM in the 6 reanalyses considered; similar correlations are observed with the other reanalyses (not shown). We believe that this level of correlation reflects differences in the definition of the indices, as opposed to errors in the reanalyses.

In the full-input reanalyses, correlation between the near surface SAM and the station-based index drops markedly in the pre-satellite period. In ERA-40 and JRA-55, the $R$ value of about 0.5 indicates that that indices share only 1/4 of the variance. The correlation, however, is effectively the same over both periods for the surface-input reanalyses. This suggest that there is greater skill in these reanalyses (and in NCEP-R1, albeit less so) that the more modern full-input reanalyses. The poor behavior of the full-input reanalyses is puzzling, but may reflect the fact that these systems have been optimized for the satellite era, and so less capable of working with more limited observations. (The surface-input reanalyses may also make use of additional historical observations that are not incorporated in the full-input reanalyses.) The consistent correlation between the SAM in the surface-input reanalyses with the station-based index suggests that we do have a good handle on near-surface variability of the SAM as far back as 1958, at least on monthly time scales. This skill likely extends through the troposphere, given the barotropic nature of annular mode variability, but is less likely to extend into the stratosphere.

As seen in Fig. 6d, the SAM in JRA-55C is better correlated with the REM in the stratosphere than it is in the troposphere during the satellite era. Similarly puzzling behavior is observed in the pre-satellite era: reanalyses which do take in measurements from the free atmosphere are better correlated in the stratosphere than below. We speculate that this is due to the broadening of the spatial structure of the SAM with height. As seen in Fig. 2a, the pattern of annular mode variability at 10 hPa extends all the way into the tropics. Similar broadening is observed at other stratospheric levels (not shown), and is even stronger at the 100 hPa level. Consequently, observations from the midlatitudes and tropics are sufficient to constrain the annular mode at higher elevations, and reanalyses do not suffer from limited observations over Antarctica.

In the Northern Hemisphere, the spatial structure of the annular mode also broadens near the tropopause, as seen with the 100 hPa NAM in Fig. 2b. In contrast to the austral hemisphere, however, the NAM becomes more localized over the pole in the mid-stratosphere: NAM variability at 10 hPa is narrower in latitude than at 850 hPa.

## 7 Representation of the annular modes in surface-input reanalyses

We explore the ability of reanalyses that only incorporate surface observations to capture annular mode variability in Figs. 8 and 9. The former compares two years of the NAM indices represented by ERA-20C with those of the REM at 1000, 100, and 10 hPa. As indicated in Fig. 6a, ERA-20C captures approximately 80% of the variability of the NAM at 100 hPa, and 60% of the variability at 10 hPa during the satellite era. The accuracy of ERA-20C appears to fluctuate from year to year. In this example, the winter of 2007-8 is captured with remarkable fidelity, while during the winter of 2006-7 there is an overall negative bias at upper levels, particularly at 10 hPa, although much of the high frequency variability is still captured.

This specific period was selected in part to highlight the ability of the ERA-20C to capture Stratospheric Sudden Warming events, marked by the yellow circles in the 10 hPa indices. ERA-20C captures the two events over this period, at least within a few days of the modern reanalyses. The annular modes in the NOAA 20CR reanalyses (not shown) are comparable to ERA-20C in the troposphere (ERA-20C doing slightly better in the Northern Hemisphere, NOAA-20CR slightly better in the Southern Hemisphere), but NOAA-20CR struggles to capture variability above the tropopause (Fig. 6). As found by Butler et al. (2017), the polar vortex in NOAA-20CR v2c is too strong (or equivalently, exhibits a cold bias), and exhibits only one major warming event between 1958 and 2011.

In Fig. 9, we compare the evolution of correlation between the annular mode indices in ERA-20C with other reanalyses that extend into the pre-satellite era. An 11 year moving window allows us to observe how the correlation changes over time. Focussing first on the boreal hemisphere, we find that the NAM at the surface (solid lines) is well constrained in all the reanalyses from 1950 on. The $R^2$ correlation with NOAA-20CR is relatively high throughout the record, albeit decaying to about 60% by the start of the century. This suggests that there may be sufficient observations in the first half of the century to capture the majority of annular variability in the troposphere.

The correlation between the ERA-20C NAM at 100 and 10 hPa with conventional reanalyses NCEP-R1 and JRA-55 appears to weaken as one moves back in time. At 10 hPa, the correlation drops noticeably around 1975. We have no means to assess the skill of NCEP-R1 before 1958. Hence the drop in correlation with ERA-20C around 1950 could be due to a loss in skill in either (or both) reanalyses. As noted earlier, the NOAA-20CR reanalyses exhibit limited skill in the stratosphere, and correlation with ERA-20C is low at all times.

In the Southern Hemisphere, the 1000 hPa annular mode indices in ERA-20C and the NOAA-20CR reanalyses are well correlated with each other as far back in time as 1950, before which point the correlation is near zero. As found from the comparison with a station-based index in Table 2, we have reason to trust tropospheric SAM indices in the surface-input reanalyses from at least the late 1950s onward. Before 1950, we likely have insufficient information to make an informed estimate. The full-input reanalyses (JRA-55 and NCEP-R1) are not well correlated with ERA-20C before 1980, but given that they diverge from the station-based SAM in the pre-satellite era, we have reason to trust ERA-20C over them. We have little reason to trust estimates of the SAM above the tropopause in any of the reanalyses in the pre-satellite era. It is interesting, however, that their appears to be non-trivial (albeit weak) correlation between the SAM in ERA-20C and those in NCEP-R1 and JRA-55 at upper levels in the 1960s, more so than observed during the 1970s.

## 8 Sampling uncertainty vs. reanalysis uncertainty

We conclude our study by focusing on the factors limiting our ability to quantify the behavior of the annular modes in our atmosphere. We contrast uncertainty associated with the finite length of the observational record, or "sampling uncertainty" with spread in the statistics between the reanalyses, or "reanalysis uncertainty". We focus exclusively on the most modern reanalyses, but the results from Section 5 suggest that nearly all the reanalyses capture variability of the annular modes fairly well during the satellite era.

### 8.1 Coupling between the stratospheric polar vortices and tropospheric jets

We first consider the uncertainty in the downward coupling between the stratosphere and troposphere associated with weak and strong vortex events, as explored by Baldwin and Dunkerton (2001). Fig. 10a,b illustrates composites of the annular mode indices as a function of height for weak vortex events in the JRA-55 reanalysis, chosen because it offers the longest record (1958-2016). Weak events, associated with Sudden Stratospheric Warmings (SSWs), and strong events are defined as in Baldwin and Dunkerton (2001): when the annular mode index at 10 hPa dips below (rises above) -3.0 (1.5) standard deviations, respectively. Additionally, a 30 day seperation is required between events. We note that these criteria largely leave out Stratospheric Final Warmings. As found by Black et al. (2006), the final break down of the vortex is often event-like, but does not project as well onto the annular mode indices in comparison with SSWs.

The extended record affords approximately twice as many events available to Baldwin and Dunkerton (2001). The key structure of the composite remains about the same, with two notable exceptions. First, following weak vortex events, the response of the tropospheric annular mode appears more abruptly, trailing by just a few days, although it still clearly lags weakening of the polar vortex in the stratosphere. The faster response at the surface is consistent with composites of hundreds of events from Coupled Model Intercomparison Project, Phase 5 (CMIP5) integrations (Charlton-Perez et al., 2013, their Fig. 5).

Second, the coupling between the troposphere and stratosphere also appears to be more simultaneous for strong vortex events, which are themselves notably less "event-like". This fact is partially obscured by the nonlinear color scale in Fig. 10a,b. In a weak vortex event, the annular mode index at 10 hPa drops 3 standard deviations – corresponding to a 1.4 km rise of the 10 hPa surface at the pole – in less than 2 weeks, the bulk of the change occurring in the final days before the event. The build up of the polar vortex is less abrupt than its destruction in SSWs – the 10 hPa index rises 1.5 standard deviations over 40 days – making the detection of strong vortex events more sensitive to the selection criteria. For example, the separation requirement plays a more important role in the definition of strong vortex events than for weak vortex events. The stronger than average vortex preceding event onset could potentially be reduced, albeit only a small amount, by requiring a longer window between events.

Fig. 10c,d shows the event-to-event standard deviation of the annular mode index as a function of height and lag. White delineates the region where the variability is near unity, equal to the climatological value. Inter-event variance is reduced at 10 hPa at the time of onset by construction, but is otherwise generally at or above the climatological variability. The mean

impact of events on the troposphere (i.e., the signal) is approximately 0.3 to 0.6 standard deviations in amplitude, and thus easily overwhelmed by this natural variability. Hence many SSWs will not appear to be associated with a downward migrating signal.

Recall that we have normalized the variance to be of order unity in the annual mean. As seen in Fig. 4, the variance is above average in the winter, when strong and weak vortex events occur. The average tropospheric value over lags -90 to 90 (or equivalently, days 0 to 90) is 1.2 standard deviations, consistent with the mean winter variability. Stratospheric variability is even more concentrated in the winter, and the standard deviation of about 2 (as seen approximately 20-40 days after strong vortex events) is comparable with the mean variability of the winter vortex.

In Fig. 11 we contrast the sampling uncertainty in these events (as determined from the JRA-55 reanalysis) with two measures of uncertainty associated with differences between the reanalyses. The sampling uncertainty (panels a,b) is expressed as a one standard deviation error bound on the composites shown in Fig. 10a,b. It is simply the event-to-event standard deviation shown in Fig. 10c,d divided by the square root of the number of events. For both weak and strong vortex events, the sampling error in the troposphere is fairly uniform in time. For weak vortex events, the average standard error over the 181 day period surrounding events is 0.21 at 300 hPa; this level is characteristic of other tropospheric levels. The average uncertainty is 0.12 for strong events, smaller because the sample size is nearly three times as large, a consequence of the weaker event threshold. We note that the event-to-event variability in the other modern reanalyses is comparable to that in JRA-55. As they provide fewer events, however, the uncertainty of the mean composite is greater, as discussed below.

For computing the "reanalysis uncertainty" we restrict the analysis to the common period 1980-2016. Fig. 11c,d shows what we have termed the "raw" uncertainty: the standard deviation between composites of weak/strong vortex constructed independently from the four modern reanalyses. At 300 hPa, the average spread between the reanalyses over the -90 to 90 day period is just 0.025 for weak events (8 times less than the sampling uncertainty) and 0.057 for strong vortex events (less than half the respective sampling uncertainty).

Much of this inter-reanalysis spread, however, arises from the fact that the composites are not necessarily built about the same events. As emphasized by Martineau et al. (2018a, their Fig. 4), differences between reanalyses are much smaller than differences between events. The threshold criteria is sensitive to subtle changes in the annular mode index, such that not all reanalyses indicate the same event dates. With respect to weak vortex events, 19 were identified in all the reanalyses, 13 of which fell on the same day across all of them. For one event, the date at which the 10 hPa NAM index crossed the -3 standard deviation threshold was offset by 1 day in the ERA-I reanalysis, and in 5 other events, the date in CFSR/CFSv2 was offset by one day.

Given the large variability of the NAM on daily time scales, these subtle offsets matter. When the event dates are fixed across all the reanalyses, the inter-reanalysis spread is further reduced, as shown in Fig. 11e,f. Here, we again plot the standard deviation between composites of weak/strong vortex events across the reanalyses, but now using identical events dates, the dates determined by the days the event occurred in the majority of the reanalysis. For weak vortex events, the average tropospheric uncertainty is reduced to 0.016, over 50% less than in the raw composite shown in Fig. 11c, and 13 times less than the sampling uncertainty.

Strong vortex "events" are less event-like by nature, leading to greater spread in their timing between the reanalyses. Analyzing each reanalysis separately, we identified 49 events in ERA-40, 52 in JRA-55 and MERRA2, and 51 in CFSR/CFSv2. The spread between these composites, Fig. 11d, thus includes some sampling uncertainty (i.e. different events), and variations due to differences in timing (which were up to a week or so in many cases).

To remove this sampling uncertainty, we focus on 53 unique events which were identified by at least 2 reanalyses. There was universal agreement on the date for 20 of these events, and three reanalyses identified the same date in 11 more cases. For the other 24 cases, at least two reanalyses differed on the timing and/or missed the event all together. When composites of weak vortex events were formed for each reanalysis using the same dates (Fig. 11f), the standard deviation between reanalyses drops substantially. At 300 hPa, the average uncertainty was just 0.011, about half the level of the smallest contour on the plot, and about 10 times less than the sampling uncertainty.

For both weak and strong vortex events, then, the uncertainty associated with differences between the reanalyses is a factor of 10 less than uncertainty associated with sampling in the extended 59 year JRA-55 record. As the sampling uncertainty decays with the square of the number of events, we would need a record that is approximately 100 times longer (i.e., 6 millenia of observations) before sampling uncertainty is reduced to the same level as the reanalysis uncertainty!

As sampling uncertainty is by far the limiting factor, the fact that high quality reanalysis of the Northern Hemisphere appears possible back to 1958 is important. The addition of 21 years lengthens the record by approximately 50% percent, thereby reducing the sampling uncertainty in the JRA-55 composite by approximately 20% percent. This is to say, if we determine the sampling uncertainty from the other reanalyses (ERA-Interim, MERRA2, or CFSR/CFSv2), the sampling uncertainty is about 20% greater, due to the reduction in the number of events. It is planned to extend ERA5 back to 1950, and we look forward to comparing this reanalysis against JRA-55 in the pre-satellite era.

## 8.2 The time scales of the annular modes

As far back as Rossby (1939) and Namias (1950), it has been recognized that there is greater persistence of the zonal mean circulation in the midlatitudes, as compared to synoptic conditions at any give location. Robinson (1996) and Lorenz and Hartmann (2001) suggest that this enhanced persistence could reflect the possibility of eddy-mean flow feedbacks. While this is clearly the case in idealized models (e.g., Garfinkel et al., 2013), more recent work by Byrne et al. (2016) questions whether such feedbacks are observed in the real atmosphere. In addition, Sheshadri and Plumb (2017) question whether the annular mode should be viewed as a stationary pattern, characterized by a single decay time scale.

A heightened sensitivity to external forcing is associated with excessive persistence of the annular modes in idealized atmosphere models (Gerber et al., 2008b) and earlier generation climate models (Kidston and Gerber, 2010; Barnes and Hartmann, 2010; Son et al., 2010), but this effect appears to be dominated by outlying models, and so less relevant to modern GCMs. Simpson and Polvani (2016) find that there is a weak correlation between the time scale of variability and jet response, but the seasonal timing is off: models with longer time scales in austral summer tend to respond more strongly in austral winter.

Capturing the correct time scales of variability has proven to be a challenging test for atmospheric models (e.g., Gerber et al., 2008a). We thus ask how well it can be constrained by the reanalyses. Baldwin et al. (2003) quantify the persistence

variations in jets by the e-folding time scale, $\tau$, of the annular mode index. It can be computed as a function of height, to contrast variability in the stratosphere and troposphere, and calendar date, to capture seasonality of the jet variations. We employ the exact same procedure as in Baldwin et al. (2003), except that we apply it to our simpler, polar-cap average definition of the annular mode. Briefly, $\tau$ is computed as the best fitting e-folding time scale to the annular mode autocorrelation function from

lag 0 to 50 days, separately at each height. It is computed separately for each day of the year, localized within the annual cycle by first applying a Gaussian filter of width 60 days at half maximum to the time series, centered about the calendar date for all years in the record.

The annular modes in the troposphere exhibit the longest time scales (Fig. 12a,b) during the seasons when the stratosphere is most variable (Fig. 4): January-February in the boreal hemisphere and November-December in the austral hemisphere. This

suggests a downward influence of the stratosphere during the seasons when coupling is most active. Gerber et al. (2010) found that models share this behavior, although both the peak in stratospheric variability and in tropospheric time scale tend to reach a maximum later in the annual cycle, particularly in the austral hemisphere.

As noted by Keeley et al. (2009), the e-folding measure $\tau$ is influenced by variability on both synoptic and interannual time scales, and so unable to comprehensively characterize the temporal variability (cf. Osprey and Ambaum, 2011). To limit the

influence of low frequency variations, here we compute the time scale on non-overlapping decadal periods, averaging them to produce Fig. 12a,b. A calculation based on the full record (thus allowing the influence of time scales greater than a decade) exhibits the same structure, but with greater $\tau$ values in regions more sensitive to low frequency variations and trends: the boreal stratosphere summer (when variability is weak) and austral spring, when stratospheric ozone can influence the flow (Thompson and Solomon, 2002).

The use of time scale calculations based on individual decades allows us to provide a rough estimate of the sampling uncertainty of $\tau$. We quantify the sampling uncertainty by the standard deviation of the $\tau$ values obtained separately for each decade. In the Southern Hemisphere we can only trust the satellite era, giving us just three samples, the minimum needed to estimate the variance. In the Northern Hemisphere, however, we can take advantage of reanalyses since 1958, giving us 5 full decades. The sampling uncertainty is substantial. We quantify it in relative terms in Fig. 12c,d, which allows us to effectively

compare both tropospheric and stratospheric values in one plot.

For the NAM, relative uncertainty in the time scale $\tau$ is between 10 and 20% over most of the annual cycle, and approaches 40% in the lower stratosphere during summer (a time, however, when the annular mode variability is quite weak). The longer time scales in the stratospheric winter, however, are comparatively robust. For the SAM (where the three decades give us only a rudimentary estimate of uncertainty), the tropospheric time scale, notably the marked increase in November and December,

still appears statistically robust. There is great uncertainty in the early spring values of $\tau$ in the stratosphere, and $\tau$ is poorly constrained throughout the atmosphere in February and March.

We contrast the sampling uncertainty in $\tau$ with the spread associated with different reanalyses in the lower panels. As with the vortex event composites, we quantify the reanalysis uncertainty by the standard deviation of $\tau$ values across the four modern reanalyses (averaged for three separate decadal calculations for which all reanalyses are available). The reanalyses agree very

well with each other, with a relative uncertainty less than 0.05 at most pressure levels and times during the year. There is some

disagreement on time scales in the boreal stratosphere during summer (when variability is weak, see also Fig. 4) and in austral stratosphere during winter. This latter case is the only instance where the reanalysis uncertainty becomes comparable to the sampling uncertainty.

The fact that sampling uncertainty generally dwarfs the reanalysis uncertainty indicates that the main limiting factor is the length of the observational record. High quality reanalysis appear to be possible in the boreal hemisphere from 1958-1978 based on our analysis in Sections 6, giving us the potential to increase the full record from approximately 40 years to 60. As discussed in the context of vortex events, a 50% increase in sample size reduces the uncertainty in the mean time scale by about 20%.

## 9 Conclusions

We have shown that the annular modes are very consistently represented by the modern reanalyses. ERA-I, JRA-55, MERRA2, and CFSR/CFSv2 can be used interchangeably for the purposes of quantifying the annular modes of our atmosphere. Preliminary analysis of ERA5 suggests equally high skill in this reanalysis as well. As detailed in Section 5, however, the other reanalyses are also quite good during the satellite era, albeit demonstrably less consistent in the austral hemisphere.

As discussed in Section 6, reanalyses of the Northern Hemisphere from 1958-1979 appears to be very consistent high into the stratosphere. The fact that the JRA-55C reanalysis (which explicitly excludes satellite observations) can match the full reanalyses in recent decades suggests that conventional observations are sufficient for a highly accurate representation of the Northern Annular Mode. In contrast, there is very little consistency in the representation of the Southern Annular Mode before satellite observations were available. In fact, near the surface, where one can estimate SAM variability on monthly time scales using station data (Marshall, 2003), the surface-input reanalyses (ERA-20C, NOAA-20CR v2 and v2c) appear superior to the full-input reanalyses. We have reason to trust the tropospheric SAM indices in the surface-input reanalyses from the 1950s onward, but little means of evaluating their representation of variability in the stratosphere prior to 1979.

In Section 8, we highlighted the fact that our ability to quantify the downward coupling and temporal variability of the annular modes is limited primarily by sampling uncertainty. This suggests that two additional decades of reanalysis have substantial value for quantifying the variability of the boreal jet stream and polar vortex. This is important for both understanding the coupling between eddies and the mean flow (e.g., Lorenz and Hartmann, 2001) and testing the fidelity of atmospheric models (e.g., Gerber et al., 2010). We encourage reanalysis centers to consider extending their modern products back until 1958 or earlier.

Finally, we have documented the ability of reanalyses constrained only by surface observations to capture variability of the annular modes in the free troposphere and stratosphere. Both NOAA-20CR and ERA-20C accurately capture annular mode variability up to the extratropical tropopause. NOAA-20CR, both versions 2 and 2c, rapidly looses the ability to track the annular modes above the tropopause. ERA-20C, however, appears to capture much of the variability high into the stratosphere, notably capturing more than 50% of the variance as high as 10 hPa for both hemispheres in recent decades.

The ability of surface-input reanalyses to constrain the stratosphere reflects the tight coupling between the troposphere and stratosphere. It should not be interpreted in a causal sense, i.e., that half of stratospheric variability is caused by surface observations. The issue of causality is important in that a number of studies have shown that modifying the stratosphere (for example, by nudging it toward observations or toward its climatology) can influence the tropospheric flow. While nudging is not equivalent to data assimilation, caution should be exercised in interpreting these experiments as one layer of the atmosphere driving another.

*Data availability.* The geopotential height data used to compute the annular modes is publicly available at the Centre for Environmental Data Analysis (Martineau, 2017) and documented in Martineau et al. (2018b). MERRA (Global Modeling and Assimilation Office, 2008) and MERRA2 (Global Modeling and Assimilation Office, 2015) output were obtained in the production of this data set.

*Competing interests.* There are no competing interest present.

*Acknowledgements.* EPG acknowledges support from the US National Science Foundation through grant AGS-1546585 to New York University. PM acknowledges support as an international research fellow of the Japan Society for the Promotion of Science. We thank Julie Arblaster and two anonymous reviewers for helpful comments on earlier versions of the manuscript, the reanalysis centers for making their data publicly available, and the Statosphere-troposphere Processes And their Role in Climate (SPARC) activity of the World Climate Research Program for their help in organizing the SPARC-Reanalysis Intercomparison Project, which inspired this research.

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

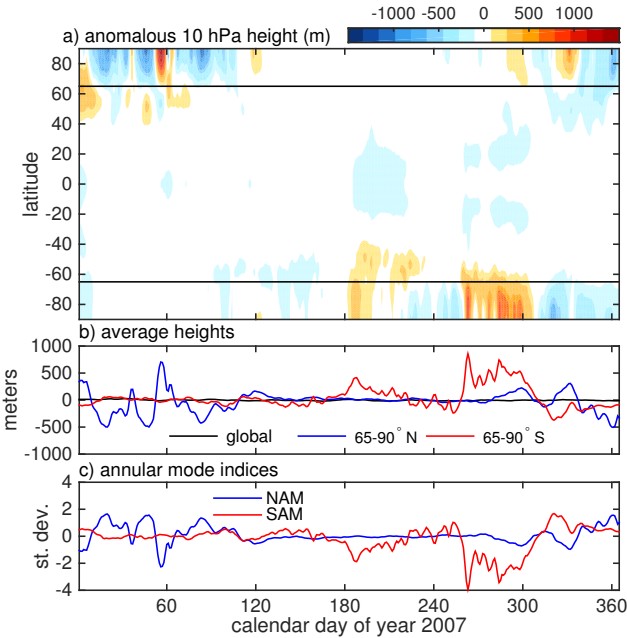

**Figure 1.** To illustrate our simplified method for calculating the annular mode indices, (a) shows anomalous zonal mean geopotential height, $\overline{Z'}$, at 10 hPa for one year, 2007. As detailed in the text, anomalies are defined relative to a smoothed annual cycle. Panel (b) shows global and polar cap averages of $Z'$ for this period, and (c) the Northern and Southern Annular Mode indices at this level, defined as -1× the respective polar cap averages less the global mean, and then normalized to have unit variance. The choice of year 2007 was arbitrary, but did exhibit a Sudden Stratospheric Warming on 24 February, which is associated with high geopotential height over the polar cap and low NAM index around day 55.

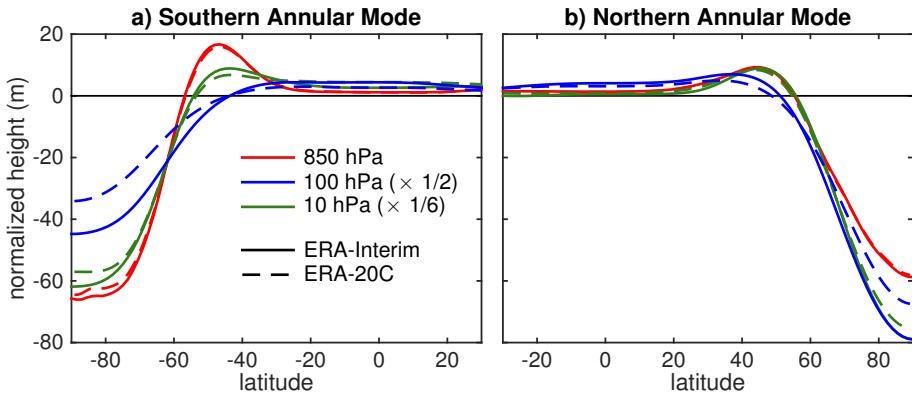

**Figure 2.** Example annular mode patterns, computed over the period 1981-2010 from ERA-Interim (solid lines) and ERA-20C (dashed lines) at three select levels. As the geopotential height anomalies increase at higher levels, the amplitude of the 100 and 10 hPa patterns has been divided by 2 and 6, respectively. The 1000 hPa patterns are nearly identical to the 850 hPa patterns; we show the latter to minimize the need for extrapolation in the austral hemisphere.

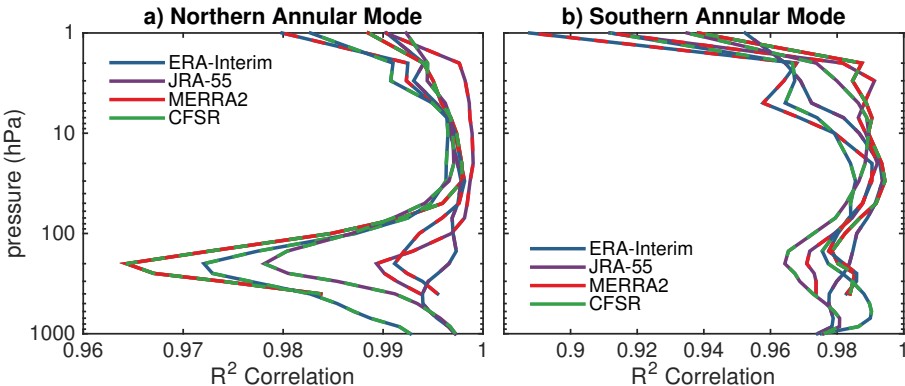

**Figure 3.** Consistency of (a) the Northern and (b) Southern Annular Mode indices in the modern reanalyses over the period 1981-2010, quantified by the pairwise correlation coefficient ($R^2$) between the annular mode indices computed from ERA-Interim, JRA-55, MERRA2, and CFSR, plotted as a function of height. The dashed colors indicate the respective indices that are being compared; for example, the green-red dashing indicates the square correlation between MERRA2 and CFSR. The square of the correlation coefficient, $R^2$, indicates the fraction of variance shared between two time series. Hence one may conclude that the NAM indices in MERRA2 and CFSR share greater than 96% of the variance at all levels.

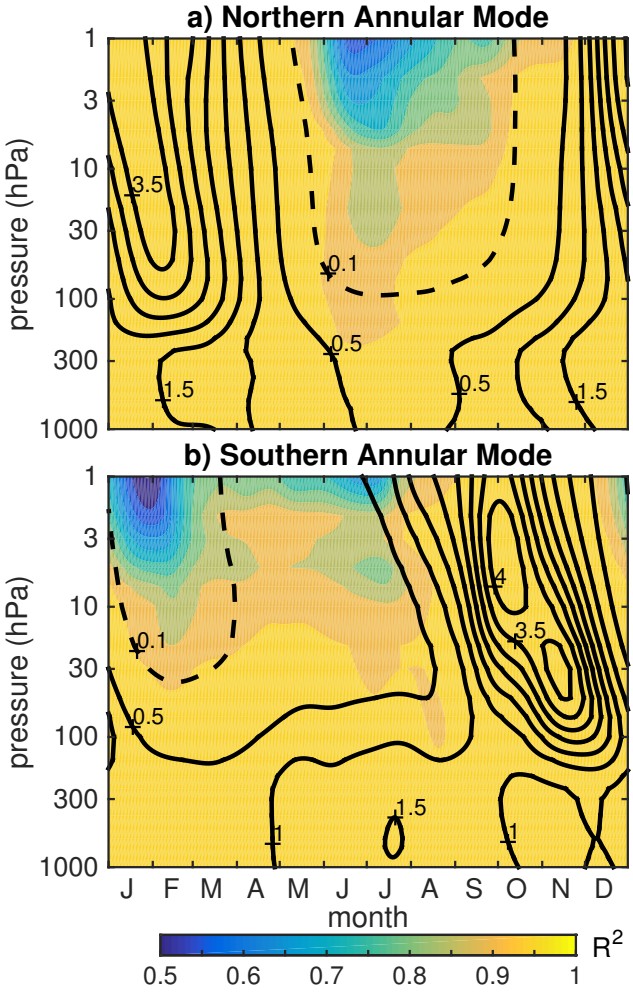

**Figure 4.** Consistency of (a) the Northern and (b) Southern Annular Mode indices in the modern reanalyses, as a function of pressure and time of year, for the period 1981 to 2010. Consistency was quantified by the average pairwise $R^2$ correlation between ERA-I, JRA-55, MERRA2, and CFSR, and indicated by the color shading. Contours indicate the variance of the REM annular mode, which has been normalized to have unit variance in the annual mean at each level; the interval is 0.5 units of variance, with an additional dashed contour at 0.1.

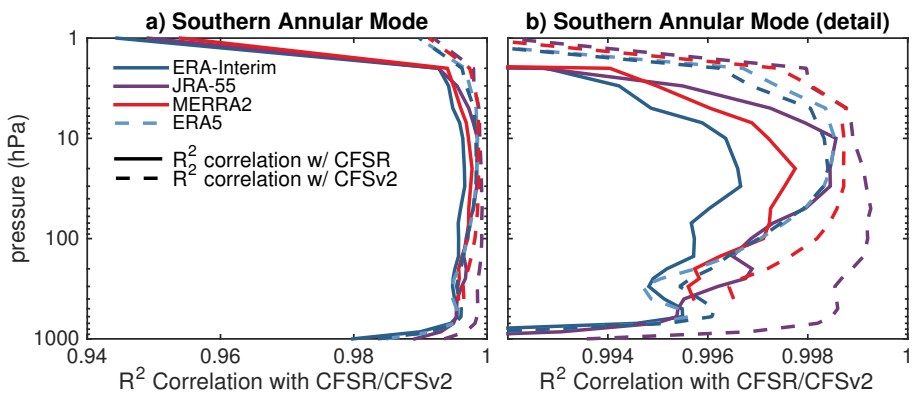

**Figure 5.** Comparison of the SAM in CFSR and CFSv2 with the other modern reanalyses. As in Fig. 3, the $R^2$ correlation is plotted: solid lines show a comparison with CFSR over the period 2005-2010, while dashed lines show a comparison with CFSv2 over the period 2011-2016. Preliminary output from the ERA5 reanalysis is available starting in 2008, and could be included in the latter comparison. We show only $R^2$ correlation with the SAM indices, as there were no discernable differences with the NAM indices between these two periods. Panel (b) shows the same as (a), but we have expanded the range of the x-axis to focus on mid-levels.

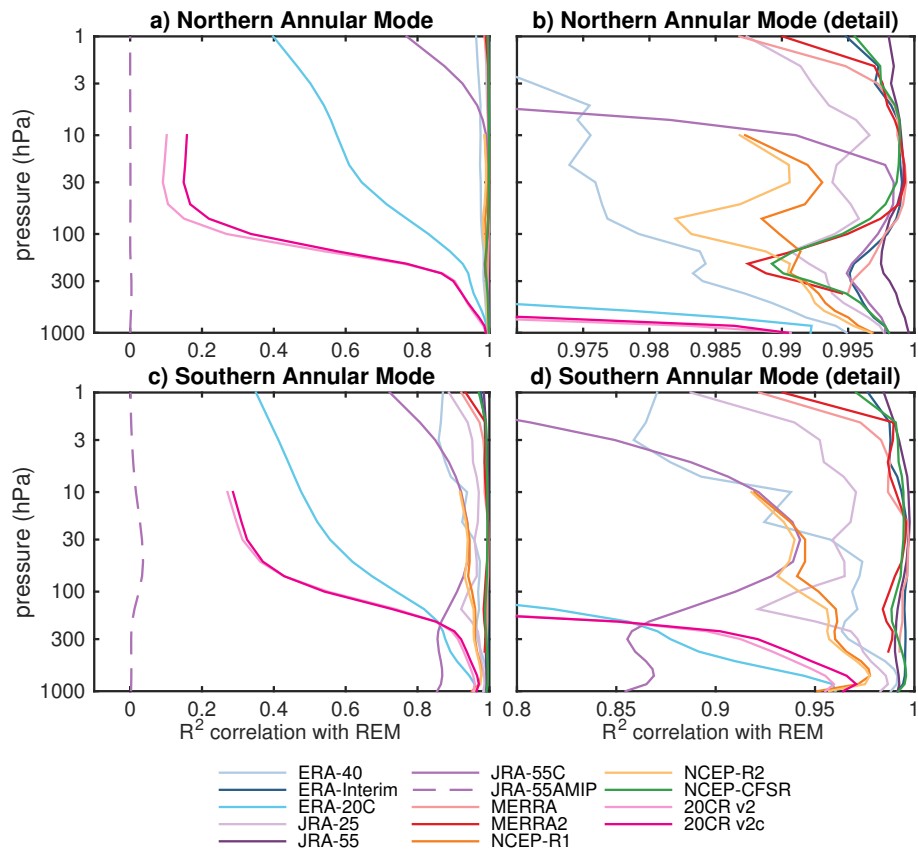

**Figure 6.** The squared correlation between the (a,b) Northern and (c,d) Southern Annular Mode indices computed from each individual reanalysis with the Reanalyses Ensemble Mean (REM). We use the standard WMO three decade climatological period, 1981-2010, for all reanalyses except ERA-40, where analysis is based on 1981-2001. Panels (b,d) show the same data, but zoomed in to highlight the more accurate reanalyses. Note the difference in the range of the abscissa: the NAM is very well constrained in all reanalyses that assimilate free atmospheric observations.

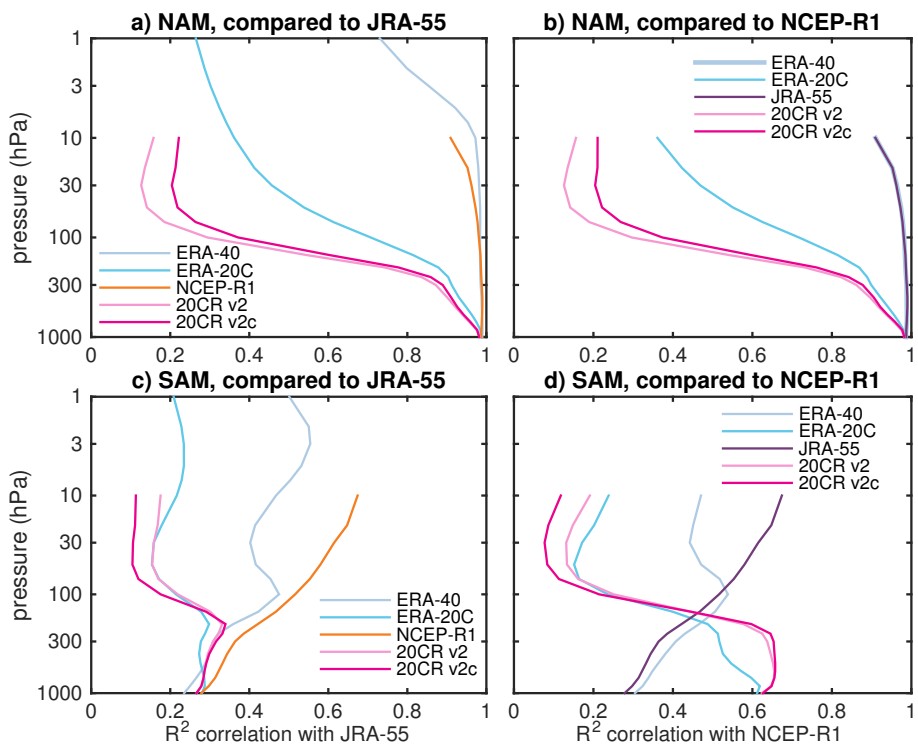

**Figure 7.** The squared correlation between (top) Northern and (bottom) Southern Annular Mode indices in various reanalyses with (left) JRA-55 and (right) NCEP-R1 during the pre-satellite era, 1958-1978.

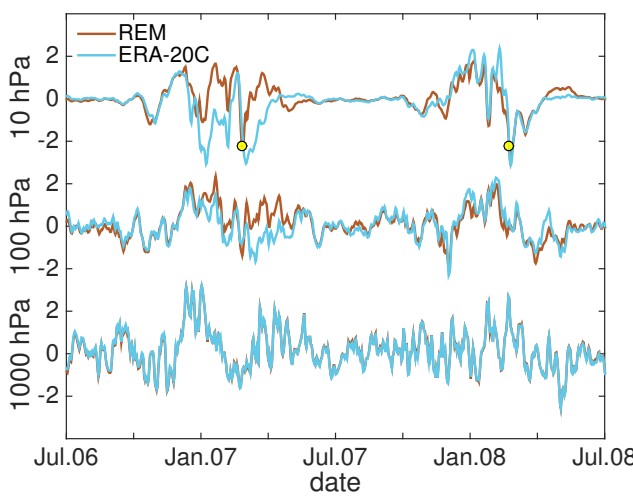

**Figure 8.** Comparison between the Northern Annular Mode timeseries from the Reanalysis Ensemble Mean (REM) and ERA-20C at 10, 100, 1000 hPa over a two year period, 1 July 2006 to 1 July 2008. Yellow dots indicate the occurrence of two Sudden Stratospheric Warming events during this period.

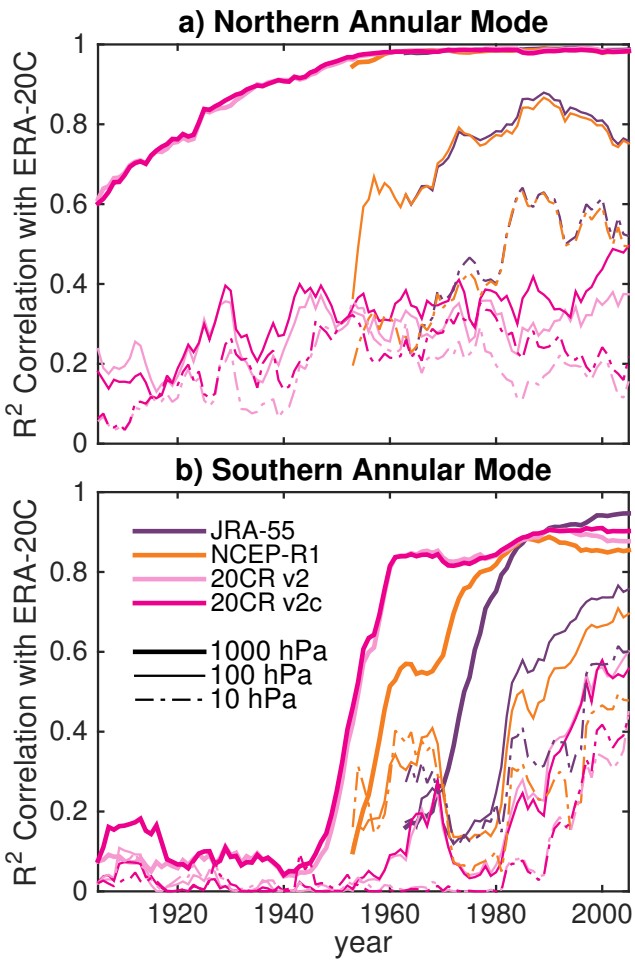

**Figure 9.** Correlation of the (top) NAM and (bottom) SAM computed from ERA-20C, as a function of time, with JRA-55, NCEP-R1, and the NOAA 20C reanalyses. The correlation is computed over a moving 11 year window, centered about the time given on the x-axis, i.e., the values at 1980 correspond to correlation between 1975 and 1985. ERA-40 was not included, as it provides comparable information to JRA-55 and NCEP-R1.

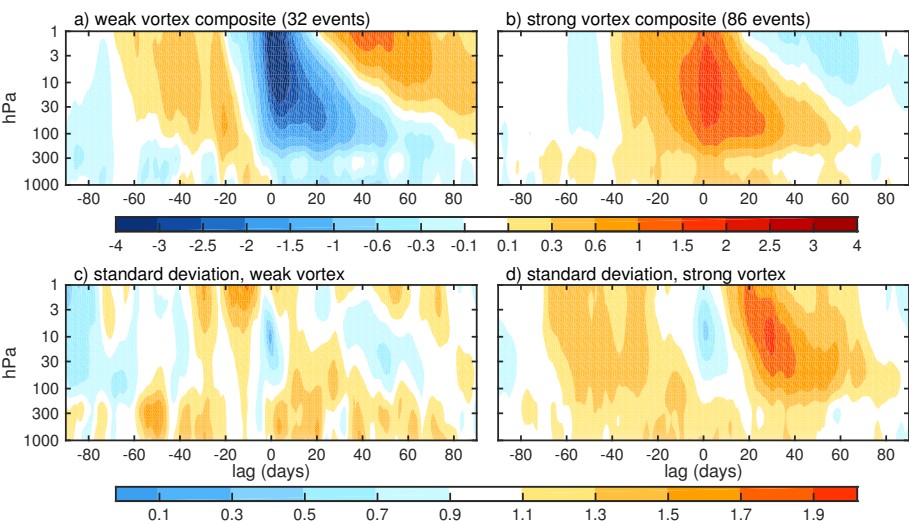

**Figure 10.** Composites of the Northern Annular Mode indices as a function of lag and pressure for (a) weak and (b) strong vortex events, based on JRA-55 reanalyses over the period 1958-2016. Following Baldwin and Dunkerton (2001), weak (strong) events are identified when the NAM index at 10 hPa drops below -3 (rises above 1.5), and must be separated by a minimum of 30 days. Panels (c) and (d) show the event-to-event standard deviation for weak and strong vortex events, respectively.

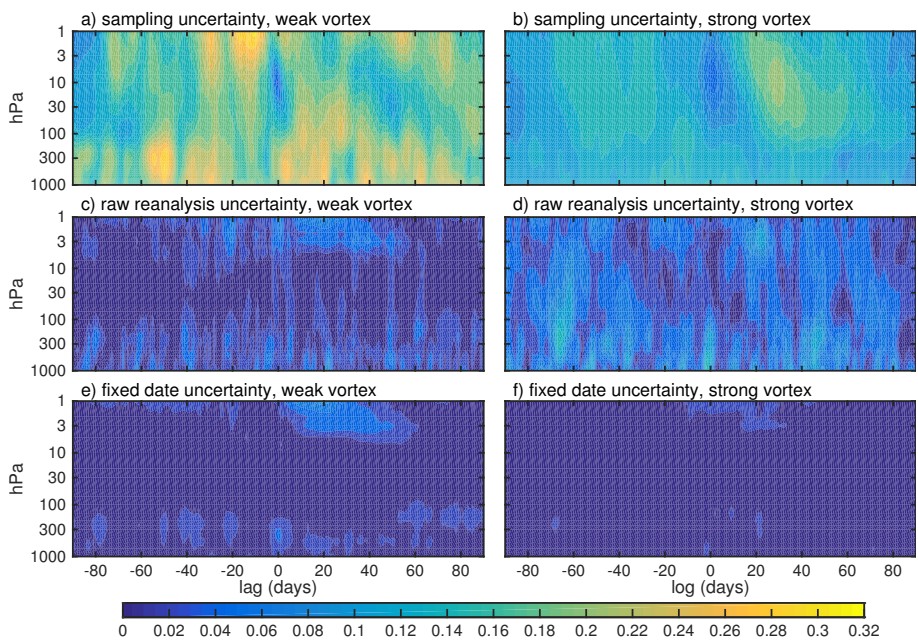

**Figure 11.** Three estimates of the uncertainty in the NAM index evolution around weak and strong events. (a) and (b) show the *sampling uncertainty* in the mean weak/strong composites shown in Fig. 10a,b, expressed as a 1 standard deviation error bound. Panels (c) and (d) show a first estimate of the *reanalysis uncertainty*: the standard deviation between composites of weak/strong vortex composites based on the 4 modern reanalyses (ERA-I, JRA-55, MERRA2, and CFSR/CFSv2) over the period 1980-2016, where events are determined independently in each reanalysis. Panels (e) and (f) show a refined estimate of the reanalysis uncertainty: the standard deviation of weak/strong vortex composites based on the 4 modern reanalyses, but now using a standardized set of event dates. See the text for further details.

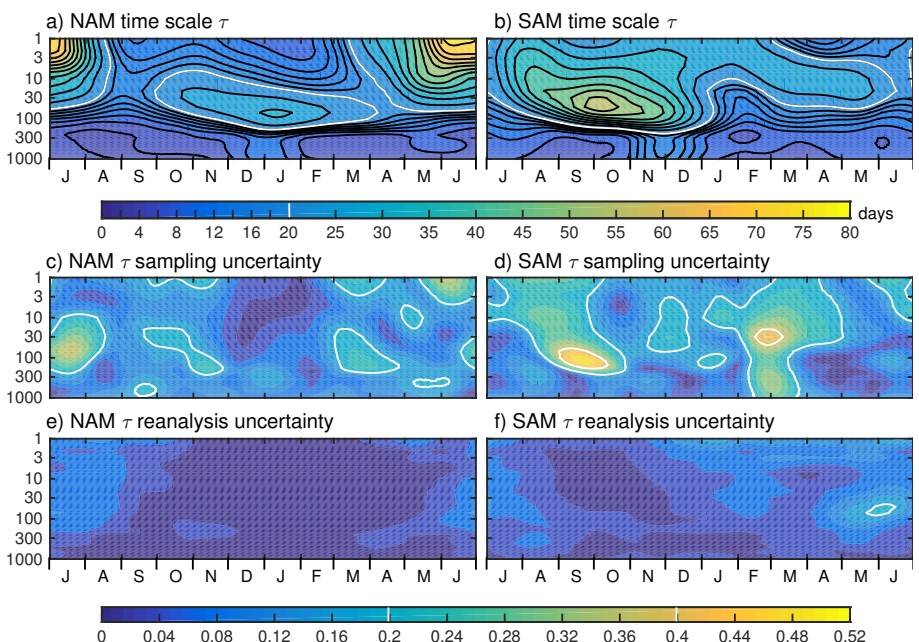

**Figure 12.** The annual cycle in the time scale of annular mode variability, $\tau$, and relative uncertainty associated with the finite record and spread between the reanalyses. As detailed in the text, the top panels show the decorrelation time scale $\tau$ of the (a) Northern and (b) Southern Annular Modes in JRA-55 analysis, based on the average of 5 and 3 independent decadal calculations (1960-9, 1970-9, ..., 2000-2009 for the NH, only the last three decades for the SH). The middle panels show the sampling uncertainty of $\tau$ for (c) the NAM and (d) the SAM: the standard deviation of the decadal calculations divided by the square root of the sample size (5 and 3, respectively). We show the uncertainty in a relative sense, normalizing by the mean time scale shown in the upper panels; otherwise errors in the troposphere (where $\tau$ is smaller) appear insignificant. The bottom panels show the reanalysis uncertainty in $\tau$ for the (e) NAM and (f) SAM: the standard deviation of $\tau$ between calculations using ERA-I, JRA-55, MERRA2, and CFSR. The standard deviation was computed across three separate decadal calculations (1980-9, 1990-9, and 2000-9) and then averaged.

**Table 1.** The atmospheric reanalyses analyzed in this study. Full-input reanalyses take into account new observations as they become available, while a conventional-input reanalysis specifically exclude satellite-based observations and surface-input reanalyses ingest only surface measurements. The availability of surface and conventional observations does fluctuate with time, however, such that these reanalyses are not immune to spurious trends associated with changes in the observation network. The JRA-55AMIP provides an free running integration of the JRA-55 atmospheric model forced with observed sea-surface temperatures, as in an Atmospheric Model Intercomparison Project (AMIP) integration.

| Reanalysis System | Coverage | Reference |
|---|---|---|
| Full-input reanalyses | | |
| ERA-40 | 9/1957-8/2002 | Uppala et al. (2005) |
| ERA-Interim | 1/1979-present | Dee et al. (2011) |
| ERA5 | 2008-2016* | — |
| JRA-25/JCDAS (JRA-25) | 1/1979-1/2014 | Onogi et al. (2007) |
| JRA-55 | 1/1958-present | Kobayashi et al. (2015) |
| MERRA | 1/1979-2/2016 | Rienecker et al. (2011) |
| MERRA-2 | 1/1980-present | Gelaro et al. (2017) |
| NCEP-R1 | 1/1948-present | Kalnay et al. (1996); Kistler et al. (2001) |
| NCEP-R2 | 1/1979-present | Kanamitsu et al. (2002) |
| CFSR | 1/1979-12/2010 | Saha et al. (2010) |
| CFSv2 | 1/2011-present | Saha et al. (2014) |
| Conventional-input reanalyses | | |
| JRA-55C | 1/1979-12/2012 | Kobayashi et al. (2014) |
| Surface-input reanalyses | | |
| ERA-20C | 1/1900-12/2010 | Poli et al. (2016) |
| NOAA-CIRES 20CR v2 (20CR v2) | 11/1869-12/2012 | Compo et al. (2011) |
| NOAA-CIRES 20CR v2c (20CR v2c) | 1/1851-12/2011 | Compo et al. (2015) |
| AMIP Simulation | | |
| JRA-55AMIP | 1/1958-12/2012 | Kobayashi et al. (2015) |

*ERA5 is currently in production, with the goal of making data available from 1950 onward.

**Table 2.** Correlation between the monthly mean Southern Annular Mode at 850 hPa with the Marshall (2003) station-based index during the pre- and post-satellite periods. Note that correlations are virtually identical if we compare with the 1000 hPa indices, as opposed to 850. While the station-based index is available from 1957 to present, we focus on these periods to optimize availability of reanalyses; in particular, ERA-40 is only available for complete years from 1958-2001.

| reanalysis | 1958-1978 | 1979-2001 |
|------------|-----------|-----------|
| ERA-40     | 0.51      | 0.87      |
| ERA-20C    | 0.83      | 0.86      |
| JRA-55     | 0.53      | 0.85      |
| NCEP-R1    | 0.70      | 0.84      |
| 20CR v2    | 0.85      | 0.85      |
| 20CR v2c   | 0.84      | 0.86      |