# Peer review of "Quantifying the variability of the annular modes: Reanalysis uncertainty versus sampling uncertainty"

_Atmospheric Chemistry and Physics, 2018_

## Referee Comment (RC1) · Anonymous Referee #1 · 4 Jul 2018

This manuscript looks at how annular modes are represented across reanalysis products, and in particular, quantifies uncertainties in sampling vs. reanalysis uncertainty. In the context of troposphere-stratosphere coupling, it would have been useful to see more Southern hemisphere analysis, particularly associated with the final warming. It seems curious that satellite observations are necessary for representing the SAM but not the NAM, one wonders why. Detailed comments that the authors might wish to consider are below.

P2 L20 The correlation between jet responses to global warming and annular mode persistence was not clear in the CMIP5 models (Simpson and Polvani 2016). In fact,

[Figure]

since the annular modes are not system modes, it is possible that the overly persistent annular more timescales in comprehensive climate models may have no implication for their response to global warming (Sheshadri and Plumb 2017). P3 L15 Please check grammar P3 L 30 Should one reasonably expect tropospheric jet variability that extends equatorward of your 65° definition (e.g. Madonna et al. 2017; Woollings and Blackburn 2012)? Fig. 3 Has this been smoothed? If so, it would be useful to people will might try to replicate this result to mention the details. P6 L18 Do you have theories as to why this would be the case? P9 L4 How do you deal with final warming (FW) events while defining strong and weak vortex events? It might be useful to examine the downward influence of FWs in both hemispheres across reanalyses, particularly since this might be of relevance to the effects of the ozone hole on tropospheric circulation in the SH. FWs are an aspect of stratosphere-troposphere coupling that have conventionally been studied using annular modes, that this manuscript completely ignores. Also, in general, stratosphere-troposphere coupling is thought to be strong at the end of winter and into early spring in the SH. P9 L19 It is somewhat inaccurate to refer to it as a downward propagating signal, as the word propagation is typically associated with the propagation of waves. Downward "influence" or "migration" might be a better choice. P11 L2 There have been follow-up studies on the eddy feedback arguments of Lorenz and Hartmann that might be worth mentioning (e.g. Byrne et al. 2016), as these indicate that the persistence of the first annular mode might not really be the right way to think of eddy feedbacks. P11 L6 Could you expand on how timescales are computed, since you do not use an EOF-based definition of the annular mode?

---

## Referee Comment (RC2) · Anonymous Referee #2 · 2 Aug 2018

Quantifying the variability of the annular modes: Reanalysis uncertainty vs. sampling uncertainty

Authors: Edwin P. Gerber and Patrick Martineau

This a very useful study in the sense that it elegantly illustrates which Reanalysis datasets and during which time periods the Northern Annular Mode (NAM) and Southern Annular (SAM) are well represented. It is not surprising that the most recent Reanalysis datasets are excellent for examining the NAM and the SAM, but it is very helpful to know that pre-satellite era data extending back to 1958 is still very useful for the NAM, but not the SAM, where the assimilation of satellite data is crucial. By

comparing various Reanalysis datasets, including those that do and do not assimilate satellite data, and those constrained only by surface pressure, the authors are able to glean much insight into what factors determine the reliability of the NAM and SAM in these datasets. This type of information is very helpful for those scientists that investigate the NAM and SAM. For this reason, I recommend that this manuscript be accepted for publication after the minor comments below are addressed. One may perhaps criticize this study for providing little insight into the physical processes that drive the NAM and SAM, but that is not the aim of the study.

Minor Comments:

1. Page 3, line 30. There have been many different definitions of the annular mode in addition to the polar cap averaged geopotential height used in this study. Baldwin and Thompson (2009) compared different annular mode definitions. In analogy with Baldwin and Thompson (2009), it would be beneficial to briefly discuss the strengths and weaknesses of the particular annular mode used in this study.

2. Figure 3. How is the consistency determined? This does not appear to be clearly defined in either the text or the figure caption. Does the consistency correspond to the average of the six pairwise correlations between the four Reanalysis datasets?

3. Page 6, line 19. What is meant by conventional observations? In the abstract, the term conventional appears to correspond to surface observations which doesn't appear to be consistent with the rest of the paragraph. Since on page 3, line 22, where it is indicated that JRA-55C data lacks satellite measurements, it appears that conventional in this paragraph corresponds to the exclusion of satellite measurements.

4. Page 7, line 6. It is stated that the NAM is consistently represented prior to the satellite era. However, this appears to be the case only for the troposphere. For the stratosphere, the ERA-20C, as indicated in the text, and the two versions of 20CR are much poorer?

5. Page 8, line 8. The relatively high R2 value between the 20CR and ERA-20C is mentioned for the early half of the 20th century. This is taken to indicate that the NAM may be reasonably accurate during this time period. It is not indicated in this paragraph that this result applies only for 1000 hPa.

6. Page 9, line 11. It is stated that the strong vortex build up is less abrupt than its decay. I don't see this in Fig. 8.

7. Page 9, lines 25-31. I did not follow how the sampling uncertainty is determined. Was the standard deviation determined for all four modern Reanalysis datasets over the lag days? A little more detail would be helpful.

8. Page 11, line 12. To remove the interannual variability, one could simply apply a high pass filter to the data with a cutoff period shorter than one year and longer than the longest e-folding time scale in the raw data.

Please also note the supplement to this comment:
https://www.atmos-chem-phys-discuss.net/acp-2018-585/acp-2018-585-RC2-supplement.pdf

---

## Author Comment (AC1) · 2 Oct 2018

Thank you for these detailed comments our manuscript. We respond below to your comments of a scientific nature, and will correct all the grammatical and typographical mistakes in our revision.

*In the context of troposphere-stratosphere coupling, it would have been useful to see more Southern hemisphere analysis, particularly associated with the final warming.*

We appreciate the author's concern about neglecting austral hemisphere. We avoided the final warming in part because several authors have noted that the response to the

Stratospheric Final Warming (SFW), is structurally different from the annular modes (Black and McDaniel 2007a,b and Sheshadri et al. 2013). In addition, the SFW is being explored as part of SPARC Reanalysis Intercomparison Project, and will be reported in Chapter 6 of the report. We had intended to highlight how the annual cycle of annular mode variability in the austral hemisphere is quite different from the boreal hemisphere, and will make an effort to better capture this in the revised draft.

*It seems curious that satellite observations are necessary for representing the SAM but not the NAM, one wonders why.*

We speculate that this is due to the sparsity of conventional observations in the Southern Hemisphere, particularly in the higher latitudes.

*P2 L20 The correlation between jet responses to global warming and annular mode persistence was not clear in the CMIP5 models (Simpson and Polvani 2016). In fact, since the annular modes are not system modes, it is possible that the overly persistent annular more timescales in comprehensive climate models may have no implication for their response to global warming (Sheshadri and Plumb 2017).*

This is a good point, and we will modify the introductory text her to be more circumspect. We believe that the link was more useful in assessing CMIP3 and CCMVal2 (the Chemistry Climate Model Validation Project, Phase 2) because there were models that more radically over-estimated the annular mode time scales.

*Should one reasonably expect tropospheric jet variability that extends equatorward of your 65ậȩ definition (e.g. Madonna et al. 2017; Woollings and Blackburn 2012)?*

Yes! As the second reviewer also had concerns about this simplified definition, we will better explain this in our revised draft. It turns out that the polar cap average does capture much of the variability of flow equatorward of 65 degrees, due to the dipole nature of the annular modes. Figure 9 of Baldwin and Dunkerton (2009) shows that correlation between the polar cap average geopotential is greater than 0.95 with the

full NAM at all levels, and greater than 0.99 in the stratosphere.

*Fig. 3 Has this been smoothed? If so, it would be useful to people will might try to replicate this result to mention the details.*

There is no smoothing, other than taking the daily mean. It looks smoother in part because chose a stratospheric level (10 hPa). At lower levels, you see more variability on synoptic scales.

*P6 L18 Do you have theories as to why this would be the case?*

This is related to the reviewers question above about why satellite measurements are needed in the austral hemisphere. We did note that JRA-55C appears to better capture the annular modes at upper levels than the surface: this may be due to the fact that annular mode captures

*P9 L4 How do you deal with final warming (FW) events while defining strong and weak vortex events?*

For consistency with previous work, we tried to follow Baldwin and Dunkerton (2001) procedure. It is possible that we have captured a few of the "dynamically" induced final warmings with our definition. In the revision, we will check to see if these late term events influence the result.

*It might be useful to examine the downward influence of FWs in both hemispheres across reanalyses, particularly since this might be of relevance to the effects of the ozone hole on tropospheric circulation in the SH.*

The influence of the ozone hole on the austral circulation will be invested as part of the SPARC Reanalysis Intercomparison Project.

*FWs are an aspect of stratosphere-troposphere coupling that have conventionally been studied using annular modes, that this manuscript completely ignores. Also, in general, stratosphere-troposphere coupling is thought to be strong at the end of winter and into*

*early spring in the SH.*

As noted earlier, we avoided the SFW in part because much of the literature (Black and McDaniel 2007a,b and Sheshadri et al. 2013), argued that the downward influence was not optimally captured by the annular modes. There is also less consistency on the precise definition of the final warming. The papers above defined it by a reversal of the winds at 50 hPa and 70 degrees latitude, while in Butler and Gerber 2018, a new definition using the winds at 10 hPa and 60 N was proposed (largely in an effort to make it more consistent with the definition of Sudden Stratospheric Warmings).

Another reasons that we avoided the final warming is that it cannot be defined from the annular mode index alone. It will be explored in the SPARC reanalysis intercomparison project. All this said, in the revision we will make a note that final warmings are important signal, especially in the austral hemisphere, where annular mode variability is concentrated at the end of the winter and spring.

*P9 L19 It is somewhat inaccurate to refer to it as a downward propagating signal, as the word propagation is typically associated with the propagation of waves. Downward "influence" or "migration" might be a better choice.*

We agree that this signal is not associated with the downward propagation of a wave. To avoid confusion, migration might be a more appropriate term and we will be careful to fix this in the revised manuscript.

*P11 L2 There have been follow-up studies on the eddy feedback arguments of Lorenz and Hartmann that might be worth mentioning (e.g. Byrne et al. 2016), as these indicate that the persistence of the first annular mode might not really be the right way to think of eddy feedbacks.*

We agree that these follow up studies should be discussed, both here and earlier in the manuscript, and will fix this in the revision.

*P11 L6 Could you expand on how timescales are computed, since you do not use an*

*EOF-based definition of the annular mode?*

We applied the same procedure as in Baldwin et al. 2003. The only difference is that we used our simplified annular mode index, as opposed to the EOF based index. In a large number of calculations that I did with CMIP5 models (which I unfortunately never published), I found that the time scales were quite similar using both indices.

---

## Author Comment (AC2) · 2 Oct 2018

Thank you for these detailed comments our manuscript. We will respond to them below, and in the revision of our manuscript.

*1. Page 3, line 30. There have been many different definitions of the annular mode in addition to the polar cap averaged geopotential height used in this study. Baldwin and Thompson (2009) compared different annular mode definitions. In analogy with Baldwin and Thompson (2009), it would be beneficial to briefly discuss the strengths and weaknesses of the particular annular mode used in this study.*

[Figure]

A practical issue for this study was handling the volume of all the reanalysis data: this simple definition in part made it possible to compare all the reanalyses with an efficient and reproducible framework. Given that there are a number of definitions of the annular mode that are effectively equivalent, we also wanted to emphasize the simplest one for future researchers.

*2. Figure 3. How is the consistency determined? This does not appear to be clearly defined in either the text or the figure caption. Does the consistency correspond to the average of the six pairwise correlations between the four Reanalysis datasets?*

Yes, by consistency, we meant to refer to the average pair-wise correlations between the four most modern reanalyses. We will clarify the figure caption in the revision.

*3. Page 6, line 19. What is meant by conventional observations? In the abstract, the term conventional appears to correspond to surface observations which doesn't appear to be consistent with the rest of the paragraph. Since on page 3, line 22, where it is indicated that JRA-55C data lacks satellite measurements, it appears that conventional in this paragraph corresponds to the exclusion of satellite measurements.*

Yes, conventional observations refers everything but satellite based measurements. In the revision, we will make this distinction more clear, especially in the abstract and section 2. In particular, we'll use more clear language, distinguishing full-input, conventional-input, and surface-input reanalyses.

*4. Page 7, line 6. It is stated that the NAM is consistently represented prior to the satellite era. However, this appears to be the case only for the troposphere. For the stratosphere, the ERA-20C, as indicated in the text, and the two versions of 20CR are much poorer?*

It was our intent to indicate that the NAM in consistent in the full-input reanalyses; for the surface-input reanalyses, there is only consistency in the troposphere. This will be clarified in the revision.

*5. Page 8, line 8. The relatively high R2 value between the 20CR and ERA-20C is mentioned for the early half of the 20th century. This is taken to indicate that the NAM may be reasonably accurate during this time period. It is not indicated in this paragraph that this result applies only for 1000 hPa.*

Yes, this refers only the surface level, and it will be clarified in the revision.

*6. Page 9, line 11. It is stated that the strong vortex build up is less abrupt than its decay. I don't see this in Fig. 8.*

We are afraid that the non-linear color bar (and potentially visual differences between the cool and warm colors) in Fig. 8 gave this incorrect impression. For weak vortex events, the index drops by over 3 standard deviations in 10-15 days, the bulk of the drop in the last 5 days. For strong vortex events, the increase in the index is only 1.5 standard deviations over approximately 40 days. We will clarify this in the revision.

*7. Page 9, lines 25-31. I did not follow how the sampling uncertainty is determined. Was the standard deviation determined for all four modern Reanalysis datasets over the lag days? A little more detail would be helpful.*

The sampling uncertainty was determined from JRA-55 alone, and quantified by the standard deviation of the composite mean in Figure 8 a,b: it is the inter-event standard deviation (shown in Fig. 8c,d) divided by the square root of the number of events. The inter-event uncertainty for the other reanalyses is comparable to JRA-55 over the satellite era (as evidenced by the reanalysis uncertainty). As there are fewer events over the satellite period, however, there is greater uncertainty in the composite mean. This will be clarified in the revised text.

*8. Page 11, line 12. To remove the interannual variability, one could simply apply a high pass filter to the data with a cutoff period shorter than one year and longer than the longest e-folding time scale in the raw data.*

The use of decadel means was chosen primarily to enable us to quantify the sampling

uncertainty. (That is, we assume that each decade was independent, and use the differences between decades as a crude measure of the sampling uncertainty.) This will be clarified in the revision.

---

## Author Response (AR1)

Dear Prof. Haynes,

We've made a number of changes to our manuscript in response to the two reviews, and feedback that we received from other members in the SPARC Reanalysis Intercomparison Project. We respond to the reviews in detail below, but wanted to highlight a few substantive changes first.

(1) It was brought to our attention that NCEP made very large changes to their reanalysis system, which came online on 1 January 2011. It is best practice to consider their reanalysis from 1979 to 2010 (NCEP CFSR) as distinct from their renalaysis from 2011 onwards, which is called NCEP CFSv2. We checked and found that this did not substantially impact the annular modes, though there is improved coherence between CFSv2 with the other modern reanalyses as compared to CFSR, which is now highlighted in a new Figure 5. To emphasize this difference between CFSR and CFSv2 to the community, we adjusted the analysis periods for Figs. 3 and 4 (previously figures 2-3).

(2) ERA5 output has become available for a limited period. We know include some preliminary assessment of this reanalysis.

(3) It was suggested that we could compare the Southern Annular Mode time series with a station-based index established by Gareth Marshall. We can only look at the monthly mean near the surface, but it did reveal that the surface-input reanalyses are superior to the full input reanalyses in the pre-satellite era. This is included in section 6 and featured in a new table 2.

(4) Both of the reviewers were concerned that our polar-cap average annular mode definition would only track variability over the pole. To clarify that it captures the full structure of the annular mode, we illustrate the associated patterns on a few select levels for two reanalyses in the new Figure 2

We thank both reviewers for their feedback on the manuscript, and believe it has improved in response to their feedback. We hope that the study is now ready for publication in ACP.

Sincerely,

Edwin Gerber and Patrick Martineau.

Detailed response to reviewer 1.

Thank you for these detailed comments our manuscript. We respond below to your comments of a scientific nature, and have corrected the grammatical and typographical mistakes from the previous draft.

*In the context of troposphere-stratosphere coupling, it would have been useful to see more Southern hemisphere analysis, particularly associated with the final warming.*

We appreciate the author's concern about neglecting austral hemisphere. We avoided the final warming in part because several authors have noted that the response to the Stratospheric Final Warming (SFW), is structurally different from the annular modes (Black and McDaniel 2007a,b and Sheshadri et al. 2013). In addition, the SFW is being explored as part of SPARC Reanalysis Intercomparison Project (S-RIP), and will be reported in Chapter 6 of the report. We had intended to highlight how the annual cycle of annular mode variability in the austral hemisphere is quite different from the boreal hemisphere, and have made an effort to better emphasize this in the revised manuscript.

Also, to better investigate the SAM, we now compare variability on monthly time scales with a station based index, established by Marshall (2003). We hope this provides a better balance of analysis of the austral hemisphere.

*It seems curious that satellite observations are necessary for representing the SAM but not the NAM, one wonders why.*

We speculate that this is due to the sparsity of conventional observations in the Southern Hemisphere, particularly in the higher latitudes.

*P2 L20 The correlation between jet responses to global warming and annular mode persistence was not clear in the CMIP5 models (Simpson and Polvani 2016). In fact, since the annular modes are not system modes, it is possible that the overly persistent annular more timescales in comprehensive climate models may have no implication for their response to global warming (Sheshadri and Plumb 2017).*

This is a good point, and we modified the introductory text here to be a bit more circumspect, and more importantly include a more comprehensive discussion of these issues in section 8.2, where both of these references are acknowledged.

We believe that the link was more useful in assessing CMIP3 and CCMVal2 (the Chemistry Climate Model Validation Project, Phase 2) because there were outlying models that more radically over-estimated the annular mode time scales.

*Should one reasonably expect tropospheric jet variability that extends equatorward of your 65◦ definition (e.g. Madonna et al. 2017; Woollings and Blackburn 2012)?*

Yes! As the second reviewer also had concerns about this simplified definition, we have expanded the discussion and include a new figure that shows how the polar-cap definition captures variability in the midlatitude (and tropics, at upper levels). The polar cap average does capture much of the variability of flow equatorward of 65 degrees, due to the dipole nature of the annular modes. Figure 9 of Baldwin and Dunkerton (2009) shows that correlation between the polar cap average geopotential is greater than 0.95 with the full NAM at all levels, and greater than 0.99 in the stratosphere.

*Fig. 3 Has this been smoothed? If so, it would be useful to people will might try to replicate this result to mention the details.*

There is no smoothing, other than taking the daily mean.  It looks smoother in part because we chose a stratospheric level (10 hPa).  At lower levels, you see more variability on synoptic scales.

*P6 L18 Do you have theories as to why this would be the case?*

This is related to the reviewers question above about why satellite measurements are needed in the austral hemisphere.  We did note that JRA-55C appears to better capture the annular modes at upper levels than the surface: this may be due to the fact that annular mode widens with height, which is now explicitly shown in Figure 2.  Thus, radiosondes from the tropics and midlatitudes have more value in constraining the annular mode.

Note that our new comparison with the Marshall (2003) station base index dives into this issue in more detail.  The surface-input reanalyses do a better job than the full-input reanalyses.  We suspect that the latter were optimized to utilize satellite measurements, and so less able to make use of surface data.  The surface-input reanalyses also make use of historic data that does not necessarily get into the full-input reanalyses.

*P9 L4 How do you deal with final warming (FW) events while defining strong and weak vortex events?*

For consistency with previous work, we tried to follow Baldwin and Dunkerton (2001) procedure.  It is possible that we have captured a few of the "dynamically" induced final warmings with our definition, which do not influence the result.  (In years where the vortex decay slowly, the annular mode index does not reach sufficiently extreme values to trigger an event.)

*It might be useful to examine the downward influence of FWs in both hemispheres across reanalyses, particularly since this might be of relevance to the effects of the ozone hole on tropospheric circulation in the SH.*

The influence of the ozone hole on the austral circulation has been investigated as part of the SPARC Reanalysis Intercomparison Project,  and will be available soon.

*FWs are an aspect of stratosphere-troposphere coupling that have conventionally been studied using annular modes, that this manuscript completely ignores. Also, in general, stratosphere-troposphere coupling is thought to be strong at the end of winter and into early spring in the SH.*

As noted earlier, we avoided the SFW in part because much of the literature (Black and McDaniel 2007a,b and Sheshadri et al. 2013), argued that the downward influence was not optimally captured by the annular modes.  There is also less consistency on the precise

definition of the final warming. The papers above defined it by a reversal of the winds at 50 hPa and 70 degrees latitude, while in Butler and Gerber 2018, a new definition using the winds at 10 hPa and 60 N was proposed (largely in an effort to make it more consistent with the definition of Sudden Stratospheric Warmings).

An issue for this paper is that all of these definitions of the final warming depend on the zonal winds, and not the annular mode index. SFWs are being explored in the SPARC reanalysis intercomparison project, and will be reported on there. All this said, in the revision we will make a note that final warmings are important signal, especially in the austral hemisphere, where annular mode variability is concentrated at the end of the winter and spring. We decided to reference Black, McDaniel, and Robinson (2006), as this is the earliest reference on these issues (to our knowledge).

*P9 L19 It is somewhat inaccurate to refer to it as a downward propagating signal, as the word propagation is typically associated with the propagation of waves. Downward "influence" or "migration" might be a better choice.*

We agree that this signal is not associated with the downward propagation of a wave. To avoid confusion, we use the term migration in the revised manuscript.

*P11 L2 There have been follow-up studies on the eddy feedback arguments of Lorenz and Hartmann that might be worth mentioning (e.g. Byrne et al. 2016), as these indicate that the persistence of the first annular mode might not really be the right way to think of eddy feedbacks.*

We agree that these follow up studies should be discussed, and have revised the introduction to section 8.2 accordingly.

*P11 L6 Could you expand on how timescales are computed, since you do not use an EOF-based definition of the annular mode?*

We applied the same procedure as in Baldwin et al. 2003. The only difference is that we used our simplified annular mode index, as opposed to the EOF based index. (In a large number of calculations that I did with CMIP5 models -- which I unfortunately never published -- I found that the time scales were quite similar using both indices.) As Baldwin et al regrettably don't really explain their method, I've tried to give the gist of it in the revised text.

Response to reviewer 2

*1. Page 3, line 30. There have been many different definitions of the annular mode in addition to the polar cap averaged geopotential height used in this study. Baldwin and Thompson (2009) compared different annular mode definitions. In analogy with Baldwin and Thompson (2009), it would be beneficial to briefly discuss the strengths and weaknesses of the particular annular mode used in this study.*

A practical issue for this study was handling the volume of all the reanalysis data: this simple definition in part made it possible to compare all the reanalyses with an efficient and reproducible framework. Given that there are a number of definitions of the annular mode that are effectively equivalent, we also wanted to emphasize the simplest one for future researchers.

To better justify this definition, we have now added a new figure (Fig. 2) which shows that the simple polar cap average definition captures the same patterns of variability as more complicated methods.

*2. Figure 3. How is the consistency determined? This does not appear to be clearly defined in either the text or the figure caption. Does the consistency correspond to the average of the six pairwise correlations between the four Reanalysis datasets?*

Yes, by consistency, we meant to refer to the average pair-wise correlations between the four most modern reanalyses. We clarified the figure caption in the revision.

*3. Page 6, line 19. What is meant by conventional observations? In the abstract, the term conventional appears to correspond to surface observations which doesn't appear to be consistent with the rest of the paragraph. Since on page 3, line 22, where it is indicated that JRA-55C data lacks satellite measurements, it appears that conventional in this paragraph corresponds to the exclusion of satellite measurements.*

Yes, conventional observations refer to everything but satellite based measurements. In the revision, will make this distinction more clear, especially in the abstract, section 2 and in the revised Table 1. In particular, we use more clear language, distinguishing full-input, conventional-input, and surface-input reanalyses.

*4. Page 7, line 6. It is stated that the NAM is consistently represented prior to the satellite era. However, this appears to be the case only for the troposphere. For the stratosphere, the ERA-20C, as indicated in the text, and the two versions of 20CR are much poorer?*

It was our intent to indicate that the NAM in consistent in the full-input reanalyses; for the surface-input reanalyses, there is only consistency in the troposphere. This has been clarified in the revision.

*5. Page 8, line 8. The relatively high R2 value between the 20CR and ERA-20C is mentioned for the early half of the 20th century. This is taken to indicate that the NAM may be reasonably accurate during this time period. It is not indicated in this paragraph that this result applies only for 1000 hPa.*

Yes, this refers only the surface level, as stated in the text.

*6. Page 9, line 11. It is stated that the strong vortex build up is less abrupt than its decay. I don't see this in Fig. 8.*

We are afraid that the non-linear color bar (and potentially visual differences between the cool and warm colors) in Fig. 8 gave this incorrect impression. For weak vortex events, the index drops by over 3 standard deviations in 10-15 days, the bulk of the drop in the last 5 days. For strong vortex events, the increase in the index is only 1.5 standard deviations over approximately 40 days. We clarify this in the revision.

*7. Page 9, lines 25-31. I did not follow how the sampling uncertainty is determined. Was the standard deviation determined for all four modern Reanalysis datasets over the lag days? A little more detail would be helpful.*

The sampling uncertainty was determined from JRA-55 alone, and quantified by the standard deviation of the composite mean in Figure 8 a,b: it is the inter-event standard deviation (shown in Fig. 8c,d) divided by the square root of the number of events. The inter-event uncertainty for the other reanalyses is comparable to JRA-55 over the satellite era (as evidenced by the reanalysis uncertainty). As there are fewer events over the satellite period, however, there is greater uncertainty in the composite mean. This is clarified in the revised text.

*8. Page 11, line 12. To remove the interannual variability, one could simply apply a high pass filter to the data with a cutoff period shorter than one year and longer than the longest e-folding time scale in the raw data.*

The use of decadal means was chosen primarily to enable us to quantify the sampling uncertainty. (That is, we assume that each decade was independent, and use the differences between decades as a crude measure of the sampling uncertainty.) This is clarified in the revision.

[revised manuscript text omitted]

---

## Author Response (AR2)

Dear Prof. Haynes,

Thank you for drawing attention to the difference between the spatial structure of the NAM and SAM in the stratosphere. I agree that I confused the issue with the reference to the "austral" hemisphere in the paragraph on lines 29-34 on p9 of the first revision of the manuscript. I also hadn't fully appreciated the differences. The annular modes in both hemispheres grow in latitudinal scale in the upper stratosphere (not discussed in this paper), but it's interesting that the 10 hPa pattern in the Northern Hemisphere is more localized than the near surface NAM.

To clarify the difference between the hemispheres, I removed "austral" in this paragraph (as it is clear that it refers to the Southern Hemisphere), and then added a short paragraph following that notes the situation is different in the Northern Hemisphere. The revised text is now lines 25-34 of page 9 in the newly submitted revision.

I did take the opportunity to catch a few more typos and reword sentences that were awkward. Also, it was brought to my attention that some satellite measurements were available before 1979. I acknowledged this is the revised text, noting that 1979 is a convenient break because it is the first full year where regular satellite measurements are available. All the changes are included in the differenced PDF included below.

Finally, in looking at the grammar, I realized that vs. (which appears in the title) is one of the many cases where Americans appear to have deviated from the Queen's English, where the abbreviation would not have the period! If possible, I thought it would best to spell out the word in the final paper.

Thank you for the careful look at our revisions, and the opportunity to further refine the paper.

Ed Gerber

[revised manuscript text omitted]